# Robust trigger wave speed in *Xenopus* cytoplasmic extracts

Jo-Hsi Huang [1,4] ✉, Yuping Chen[1,3,4], William Y. C. Huang[1], Saman Tabatabaee [1] & James E. Ferrell Jr [1,2] ✉

Self-regenerating trigger waves can spread rapidly through the crowded cytoplasm without diminishing in amplitude or speed, providing consistent, reliable, long-range communication. The macromolecular concentration of the cytoplasm varies in response to physiological and environmental fluctuations, raising the question of how or if trigger waves can robustly operate in the face of such fluctuations. Using *Xenopus* extracts, we find that mitotic and apoptotic trigger wave speeds are remarkably invariant. We derive a model that accounts for this robustness and for the eventual slowing at extremely high and low cytoplasmic concentrations. The model implies that the positive and negative effects of cytoplasmic concentration (increased reactant concentration vs. increased viscosity) are nearly precisely balanced. Accordingly, artificially maintaining a constant cytoplasmic viscosity during dilution abrogates this robustness. The robustness in trigger wave speeds may contribute to the reliability of the extremely rapid embryonic cell cycle.

Frog eggs are large cells that are particularly well-suited to quantitative biochemical studies. The eggs are about 1.3 mm in diameter and 1 μL in volume, which makes them amenable to single-cell biochemical assays[1]. Moreover, they can be lysed with minimal dilution, and the undiluted cytoplasm can be recovered and studied[2,3]. These egg extracts self-organize into cell-like compartments[4], and like the cells from which they are derived, they can carry out rapid cell cycles[2,5,6] and, under adverse conditions, die by apoptosis[7,8]. Indeed, *Xenopus* egg extracts have provided important insights into the regulation of both the cell cycle and apoptosis.

The large size of the frog egg presents a challenge shared by other large cells and tissues: how to coordinate rapid processes like mitotic entry and apoptotic death across such large distances. Early modeling work on the cell cycle suggested that mitosis might spread through the egg via trigger waves of Cdk1 activity[9]. Trigger waves can occur in systems with positive feedback loops, and they spread faster over large distances than diffusion alone would allow[10,11]. Experimental work has shown that mitosis does spread through *Xenopus* cytoplasm via trigger waves[5,12], at a speed of ~60 μm min⁻¹, and apoptosis does as well, at

about half that speed[8]. A growing body of evidence suggests that trigger waves may be a common way of transmitting signals over large distances in biological systems. Action potentials and calcium waves are familiar examples of trigger waves, as are intercellular cAMP waves in swarming *Dictyostelium*[13–15] and intercellular ERK waves in wounded fish scales[16] and mouse skin[17]. Recent work suggests that the remarkable regeneration of an amputated planarian depends upon signals transmitted from the wound site via intercellular trigger waves of ERK activation[18].

The cytoplasm is a crowded, spatially organized mixture of organelles, macromolecules, and small molecules. Protein concentrations in *Xenopus* extracts[19–21] and mammalian cell lines[22] are typically on the order of 75 mg mL⁻¹, although it is higher in some cells, e.g. erythrocytes. It has been conjectured that the nominal cytoplasmic protein concentration maximizes the speed of important biochemical processes[23]. The extent to which this conjecture holds true for cells awaits experimental investigation. Conversely, protein concentration is dynamic; it falls by ~15% when cells enter mitosis[24–26], and by up to 50% when cells undergo senescence[22,26,27] and when chondrocytes

[1]Department of Chemical and Systems Biology, Stanford University School of Medicine, Stanford, CA 94305, USA. [2]Department of Biochemistry, Stanford University School of Medicine, Stanford, CA 94305, USA. [3]Present address: Shenzhen Institute of Synthetic Biology, Shenzhen Institutes of Advanced Technology, Chinese Academy of Sciences, Shenzhen, China. [4]These authors contributed equally: Jo-Hsi Huang, Yuping Chen. ✉e-mail: johsi@stanford.edu; james.ferrell@stanford.edu

differentiate[28]. Cells change in volume when attaching to substrates of different stiffnesses[26,29,30], and recent work indicates that neutrophils swell by ~15% in response to chemoattractants, and that the swelling facilitates rapid migration[31]. The extent to which changes in volume, and changes in cytoplasmic concentration, impact the biochemistry of living cells is as yet poorly understood.

Here we ask how long-range communication via trigger waves is affected by changes in the concentration of cytoplasmic *Xenopus* egg extracts. We show that both mitotic and apoptotic trigger waves can be generated and propagated over a wide range of cytoplasmic concentrations. The wave speeds are maximal or near maximal at a 1 x cytoplasmic concentration, in line with Dill's conjecture that the nominal 1 x concentration maximizes the speeds of critical biochemical processes[21,23], and in the case of apoptotic trigger waves the speed is almost invariant over concentrations from 0.1 x–2 x. We derive a simple general equation for trigger wave speed as a function of cytoplasmic concentration, which shows how balanced opposing effects are responsible for this robustness, and show that the equation satisfactorily accounts for our experimental observations. Finally, we show that disrupting the balance by maintaining a constant viscosity when diluting the extracts makes trigger wave speed highly sensitive to cytoplasmic concentration.

## Results

### Mitotic trigger waves in concentrated and diluted extracts

Mitosis is brought about by a complex, interconnected regulatory system centered on a protein kinase, cyclin B-Cdk1, and two opposing phosphatases, PP1 and PP2A-B55 (Fig. 1a). Several positive feedback and double-negative feedback loops are embedded in this regulatory system; for example, active cyclin B-Cdk1 turns on its activator Cdc25, and cyclin B-Cdk1 and PP2A-B55 antagonize each other via intertwined double-negative feedback loops (Fig. 1a). The net result of these feedback loops is that the system functions as a bistable switch[32,33], and this bistability is key for the propagation of the mitotic state as a trigger wave[12].

To see how robust mitotic trigger waves are to changes in the concentration of the cytoplasm, we began by making either a 1 x cytoplasmic extract or a concentrated extract on a Microcon spin column. From 4 independent preparations, the 1 x cytoplasmic protein concentration was $57.9 \pm 3.4$ mg mL$^{-1}$ (mean ± S.E.M., $n = 4$; Fig. 1c), in line with other estimates[19–21], and not far from the protein concentrations measured for three common mammalian cell lines (~75 mg mL$^{-1}$)[22]. The concentrated extract was $116 \pm 6.0$ mg mL$^{-1}$ (mean ± S.E.M., $n = 6$; Fig. 1c); hereafter we will refer to it as a 2 x retentate. The flow-through, which we will refer to as the filtrate, from the spin column had a protein concentration of less than 0.01 mg mL$^{-1}$ (Fig. 1c).

We then diluted the 1 x extract or 2 x retentate to various extents. In other recent work we used filtrate for the dilutions[21]. Here we have used XB buffer without sucrose rather than filtrate, which allowed us to produce larger volumes of diluted extracts, and we verified that the behaviors of the buffer-diluted and filtrate-diluted extracts were similar (Supplementary Fig. 1).

We added demembranated sperms and SiR-tubulin to the extracts and dilution buffers, made the dilutions, and aspirated extracts into ~100 μm or ~300 μm inside-diameter polytetrafluorethylene (PTFE) tubes under gentle vacuum. The tubes were placed under mineral oil and followed by fluorescence video microscopy. Figure 1d, left panel, shows a typical result. At the first time point shown here, the extract was in interphase with stable microtubules throughout the length of the tube. Within a few minutes, mitosis began near the bottom of the tube and at a locus about 4 mm up the tube. As judged by the depolymerization of the fluorescent interphase microtubules, mitosis spread outward from these two loci in a linear fashion (Fig. 1d). Mitotic exit followed about 12 min after mitotic entrance, and it also spread

linearly outward from the same two locations. Figure 1d, right panel, shows the same data where instead of imaging the whole tube, we recorded SiR-tubulin fluorescence intensity along a line down the middle of the tube and then assembled the data into a kymograph. In either representation, the trigger wave character of mitotic propagation is apparent, and the speed of the mitotic front was $60.2 \pm 3.2$ μm min$^{-1}$ (mean ± S.E.M., $n = 9$), similar to previously reported mitotic wave speeds[5,12,34].

Next we examined how the cell cycle period and the speed of the mitotic waves were affected by changes in cytoplasmic concentration. Figure 1e shows examples of kymographs from a diluted 1 x extract and diluted 2 x retentate. The periods of the first cycles and the wave speeds were calculated and are summarized in Figs. 1f and 1g, which include multiple experiments and more dilutions. Several general trends are apparent. First, the wave speeds were similar for 1 x extracts and 2 x retentates diluted back to 1 x, but the periods were different, with the diluted 2 x retentates having longer cell cycle periods than the corresponding 1 x and diluted 1 x extracts. Second, the cell cycle periods tended to be longer in diluted extracts than in concentrated extracts (Fig. 1f). Third, the diluted extracts tended to live longer than the concentrated extracts; a wave of apoptosis, which destroys the microtubule fluorescence, can be seen in the second and in fourth kymographs (Fig. 1e). Fourth, the most concentrated extracts tended to arrest in mitosis with depolymerized microtubules (Supplementary Fig. 2; cf. Figure 1e, where the extract did not arrest in mitosis). Fifth, mitosis tends to occur approximately synchronously at first and then transitions to spatially ordered waves, as previously noted[5,12,35]. And finally, the speeds of the mitotic waves were relatively invariant, with only the extracts at greater than 1 x showing some slowing of the waves. Diluting the extract below 1 x slightly increased the wave speed (by ~10%). For comparison, if the wave speed were determined by a bimolecular reaction, decreasing the extract from 1 x–0.5x might be expected to decrease the wave speed by 75%. Both the cell cycle frequency and mitotic wave speed were at or near their maximal values at 1 x cytoplasmic concentration, consistent with Dill's conjecture[23].

### Apoptotic trigger waves in concentrated and diluted extracts

Apoptosis is mediated by a complex system of regulators that bring about the activation of caspases 3 and 7, so-called executioner caspases that cleave diverse cellular proteins and bring a halt to the basic processes of life (Fig. 2a). There are several potential positive feedback loops in the apoptotic control system (Fig. 2a), raising the possibility that caspase activation could spread via trigger waves. In many cell types, apoptosis spreads through the cytoplasm in a wave-like manner[36–38], and in *Xenopus* egg extracts, where it is easy to obtain length scales over which the distinction between diffusive spread and trigger wave spread is unambiguous, it is clear that the fronts of caspase activation represent trigger waves that propagate without slowing down or decreasing in amplitude[8]. The manipulability of extracts allowed us to assess the sensitivity of the apoptotic trigger wave speed in such extracts to cytoplasmic concentration, and to tease out the contributing effects quantitatively.

Interphase *Xenopus* egg extracts were prepared and were mixed with a rhodamine-based fluorogenic sensor of caspase 3/7 activation, (Z-DEVD)$_2$-R110, and a proteasome inhibitor (MG-132), which decreased the background level of R110 fluorescence and hence improved the signal-to-noise ratio of the experiment. The extracts were then loaded into thin (~100 μm diameter) PTFE tubes (Fig. 2b). Apoptosis was induced by briefly dipping one end of the tube into a reservoir of apoptotic extract, prepared by adding cytochrome c (2 μM) to fresh extract and incubating at room temperature for 30 min. The induced tubes were then immersed in heavy mineral oil in custommade imaging chambers and imaged at 2 min intervals at room temperature.

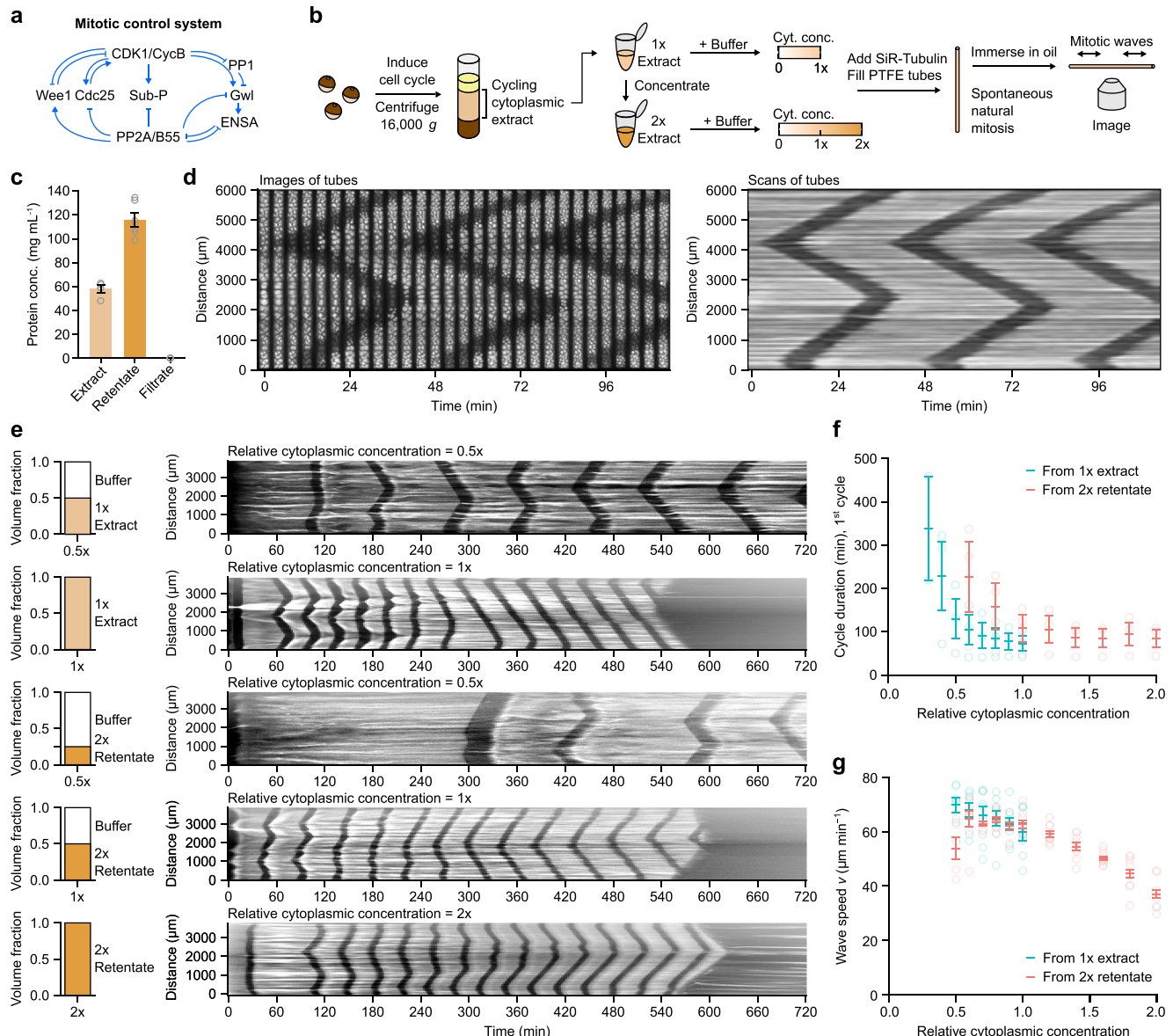

**Fig. 1 | Mitotic trigger wave speed is robust to change in cytoplasmic concentration. a** Schematic view of the mitotic control network. Note the multiple interconnected positive and double-negative feedback loops. **b** Preparation of cycling *Xenopus* egg extracts and the workflow of monitoring spontaneous mitotic trigger waves in thin PTFE tubes by epifluorescence microscopy. **c** Measurements of protein concentrations in the original and concentrated extract (retentate) as well the filtrate from the ultrafiltration filters. *n* = 4 for extracts, 6 for retentates, and 5 for filtrates. **d** A montage (left) and its corresponding kymograph (right) of a single tube undergoing 3 rounds of mitosis in the course of ~2 h. The bright signal is polymerized microtubules stained with SiR-tubulin and the dark bands correspond

to the mitotic state in which most microtubules are depolymerized. **e** Representative kymographs (right column) of extracts of different cytoplasmic concentrations prepared from 1 x extracts (top 2 rows) or from 2 x retentate (bottom 3 rows). Left column shows the volume fractions of buffer (XB buffer without sucrose) and extracts that went into the samples. **f** Duration of the first completely observable cell cycle under the microscope, starting from interphase. Means ± S.E.M. are shown, *n* = 3 (three independent experiments). **g** Speeds of mitotic trigger waves at different cytoplasmic concentrations. Means ± S.E.M. are shown. Data is compiled from *n* = 8 independent experiments. Source data are provided as a Source Data file.

Figure 2b shows the results of a typical experiment. Apoptosis, as detected by bright R110 fluorescence, first initiated at the dipped end of the tube and then spread toward the other end. The propagation speed in this experiment was 29.6 μm min⁻¹; average speeds from 25 independent experiments were 27.5 ± 0.8 μm min⁻¹ (mean ± S.E.M.). This is similar to the speeds seen in the cycling extracts that underwent apoptosis in Fig. 1e (28.1 and 27.7 μm min⁻¹) and agree well with previous reports[8].

We then altered the cytoplasmic concentration of the egg extract by diluting either a 1 x extract or a 2 x retentate. We verified apoptotic wave speed responds similarly to 3 different diluents (Supplementary

Fig. 3) and chose XB buffer without sucrose as the primary diluent for further experiments. Figure 2c shows kymographs of R110 fluorescence as a function of time for the original 1 x extract and a 0.5 x obtained by dilution, and for a 2 x extract and a 1 x extract reconstituted from the 2 x extract by dilution; Fig. 2d shows data from 9 independent experiments, including additional extract concentrations. Overall, the apoptotic wave speed was almost invariant (Fig. 2d). There was no measurable change over a concentration range of 0.5x to 1x, and the speed decreased by only about 15% as the concentration increased to 2 x (Fig. 2d). Thus, apoptotic wave speed is highly robust to variations in cytoplasmic concentration.

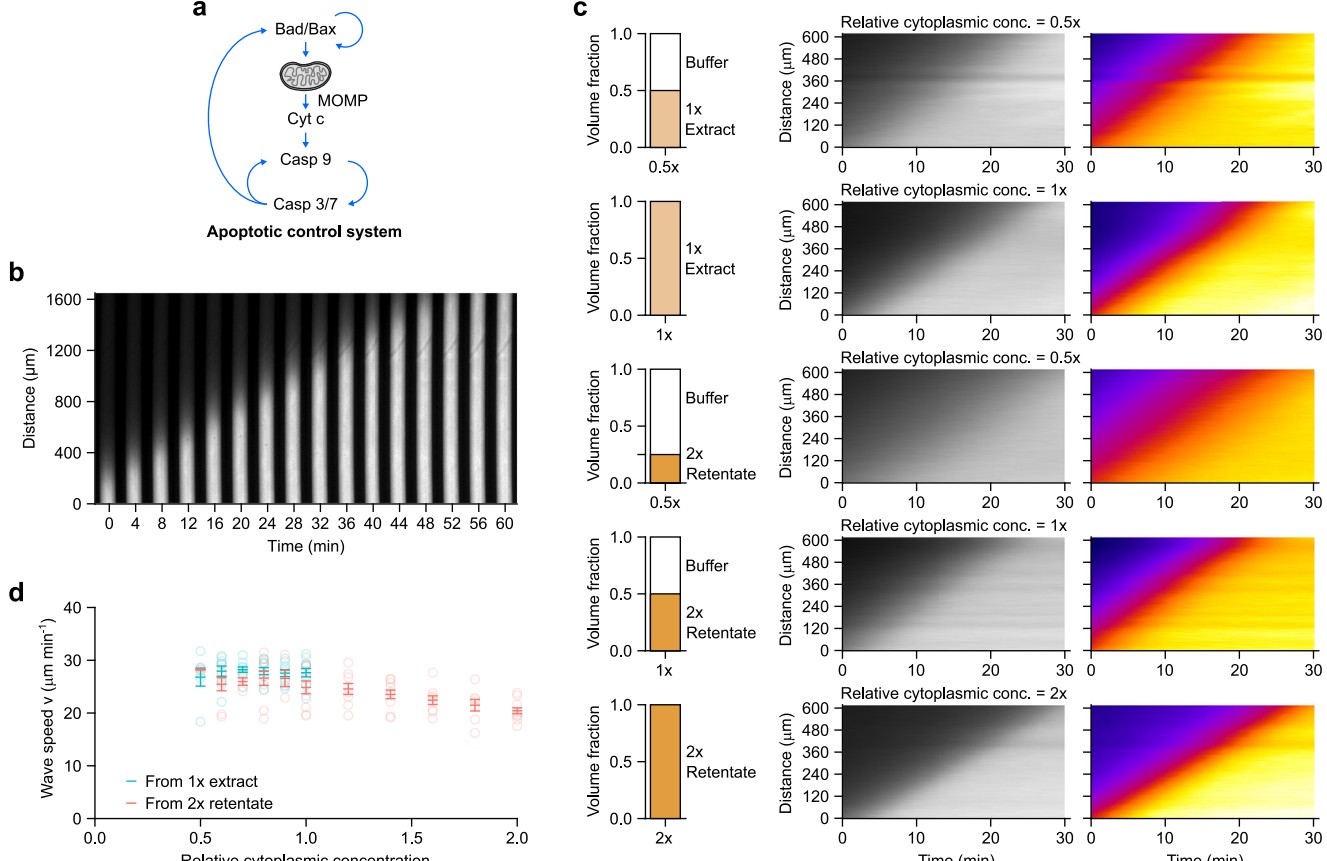

**Fig. 2 | Apoptotic trigger wave speed is robust to change in cytoplasmic concentration. a** Schematic view of the apoptotic control system. Note the multiple positive feedback loops. **b** A representative montage of an apoptotic trigger wave in a PTFE tube induced from the lower end. Bright signal is rhodamine 110 released by caspase 3/7 cleavage of (Z-DEVD)$_2$-R110, which reports the activation of caspase 3/7. **c** Representative kymographs (middle and right columns) of extracts at different cytoplasmic concentrations prepared from either 1 x extract (top 2 rows) or retentate (bottom 3 rows). Rhodamine 110 fluorescence is shown in gray scale (middle column) or pseudo-color (right column). Pseudo-coloring demonstrates that a range of fluorescence thresholds would give very similar estimates of trigger wave velocity. **d** Speeds of apoptotic trigger waves at different cytoplasmic concentrations. Means ± S.E.M. are shown. Data are compiled from $n = 9$ independent experiments. Source data are provided as a Source Data file.

## Deriving an expression for wave speed as a function of cytoplasmic concentration

To try to understand these trends, and in particular to see how wave speed can be so insensitive to cytoplasmic concentration, we derived an expression for wave speed as a function of cytoplasmic concentration. We began with Luther's formula[39,40], which relates the speed of trigger wave propagation to its underlying reaction and diffusion dynamics:

$$v = 2\sqrt{\kappa D}. \quad (1)$$

The parameter $\kappa$ is the apparent first-order rate constant (in units of time$^{-1}$) for the positive feedback and $D$ is the diffusion coefficient for the active enzymes. In principle, a number of individual kinetic parameters and component concentrations could contribute to $\kappa$; here we will assume that the overall dynamics of the system can be approximated by logistic growth:

$$C_{active} = \frac{C_{total}}{1 + \left(\frac{C_{total} - C_o}{C_o}\right)e^{-\kappa t}} = \frac{C_{total}}{1 + \left(\frac{C_{total} - C_o}{C_o}\right)e^{-kC_{total}t}}, \quad (2)$$

where $C_{active}$ is the concentration of the active enzyme (e.g. Cdk1 for mitotic waves and Caspase 3/7 for apoptotic waves), $C_{total}$ is the total concentration of $C$, $C_o \equiv C_{active}$ at $t = 0$ is the initial concentration of $C_{active}$, and the parameter $k$ is the effective second order rate constant (in units of concentration$^{-1}$ time$^{-1}$) for the activation of $C$. To assess the

reasonableness of this assumption, we fitted Eq. 2 to a published detailed time course for Cdk1 activation in cycling *Xenopus* extracts[41]. As shown in Fig. 3a, the time course was well approximated by a logistic function until Cdk1 activity began dropping during late mitosis. This drop is due to the activation of the APC/C and the destruction of cyclins, which causes the cycling system to exit mitosis. We also determined whether the logistic equation approximated the dynamics of caspase 3/7 activation. The time course of production of caspase probe fluorescence at an arbitrarily chosen position in the Teflon tube (Fig. 3b) was sigmoidal (Fig. 3c). We then used an ordinary differential equation model relating the time course of fluorescence production to the underlying time course of caspase activation (see Methods). The resulting curves (Fig. 3b, c, red curves) could be fitted well to the measured data in Fig. 3b and to the inferred activities in Fig. 3c, especially during the initial rise in activity.

As a final check, we calculated estimated trigger wave speeds using Luther's equation (Eq. 1) and the inferred values of $\kappa$ from Fig. 3a and c and an assumed diffusion coefficient of 15 μm$^2$ s$^{-1}$ for the mediators of the trigger wave spread (see below). The resulting inferred speeds were 50 μm min$^{-1}$ for mitotic waves and 28 μm min$^{-1}$ for apoptotic waves, in excellent agreement with the experimentally measured speeds (60.2 ± 3.2 and 27.5 ± 0.8 μm min$^{-1}$; see above), and within the range of speeds expected given the uncertainty in the relevant diffusion coefficients (see below). Thus, the applicability of Luther's formula and the assumption that the dynamics of the positive feedback are described by the logistic equation are justified.

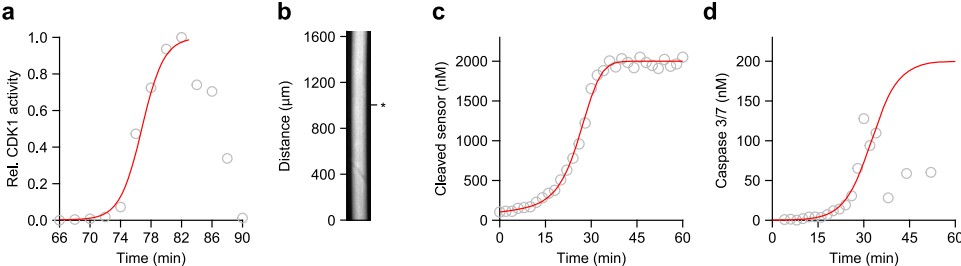

**Fig. 3 | Mitotic and apoptotic activities are well-approximate by the logistic equation. a** Relative CDK1 activity as a function of time at mitotic onset, normalized to a maximal activity of 1, as measured by H1 kinase activity assay, can be well-fitted by a logistic model. Data are taken from Pomerening et al.[41]. **b** Apoptosis as a function of time, measured at one point in a Teflon tube of extract. Fluorescence was followed as a function of time by microscopy. **c** Kinetics of (Z-DEVD)$_2$-R110 cleavage. Open circles are data at each time point. The red curve is based on the ODE model for caspase 3/7 activation and (Z-DEVD)$_2$-R110 cleavage (Eq. 16) fitted to the data. **d** Kinetics of caspase 3/7 activation. Open circles are concentrations of active caspase 3/7 calculated based on the ODE model (Eq. 15). The red curve is the logistic equation fitted to the data. Source data are provided as a Source Data file.

We may rewrite Eq. 1 as:

$$v = 2\sqrt{kC_{total}D},\tag{3}$$

Note that all three of the variables under the square root sign might vary with the overall cytoplasmic concentration $\phi$. This can be expressed as:

$$v(\phi) = 2\sqrt{k(\phi)C_{total}(\phi)D(\phi)}\tag{4}$$

We assume that $C_{total}(\phi)$ is simply proportional to the overall cytoplasmic concentration $\phi$:

$$C_{total}(\phi) = \phi C_1,\tag{5}$$

where $C_1$ denotes total caspase 3/7 or Cdk1 concentration at 1x cytoplasmic concentration. This assumption leaves two functions, $D(\phi)$ and $k(\phi)$, to be experimentally determined.

For mitotic trigger waves, likely mediators of the spatial spread include cyclin B-Cdk1, Cdc25, and Gwl, proteins with molecular weights of ~100 kDa. For apoptosis, plausible mediators include activated caspase 3 and 7 heterotetramers (~60 kDa) and cytochrome c (12 kDa). Therefore, we chose a similarly-sized probe, Alexa Fluor 488-labeled bovine serum albumin (AF488-BSA; ~67 kDa), for diffusion measurements. We measured its diffusion coefficient as a function of cytoplasmic concentration by fluorescence correlation spectroscopy (FCS). The diffusion mode of AF488-BSA was fairly close to Brownian in XB buffer without sucrose ($\alpha \approx 0.9$) and more subdiffusive in extracts ($\alpha \leq 0.8$; Fig. 4a, Supplementary Fig. 4). These observations are consistent with previous reports[42–45]. The effective diffusion coefficient $D_{eff}(\phi)$, which is calculated assuming Brownian motion rather than subdiffusive motion, decreased exponentially with cytoplasmic concentration ($R^2 = 0.953$; Fig. 4b), as predicted by Phillies' law[46,47]. The fitted effective diffusion coefficients in XB buffer without sucrose ($\phi = 0$) and 1x extract ($\phi = 1$) were 32 and 15 $\mu m^2 \, s^{-1}$ (Fig. 4b), respectively, again consistent with previous measurements[42]. We can therefore express the scaling of the effective diffusion coefficient $D_{eff}(\phi)$ as:

$$D_{eff}(\phi) = D_0 e^{-g_D \phi}\tag{6}$$

where $D_0$ is $D_{eff}$ in buffer (i.e. at a cytoplasmic concentration of 0) and $g_D$ is a dimensionless scaling factor with a fitted value of 0.765. If our assumption of the molecular mass of the trigger wave mediator is off by a factor of 3, the value of $D_0$ could be off by as much as 1.5-fold (21–48 $\mu m^2 \, s^{-1}$).

The remaining contributor to Eq. 4 is $k(\phi)$, the effective second order rate constant for the positive feedback reactions. We focused on

the $k(\phi)$ function for apoptosis rather than mitosis because we had a more direct probe for caspase activity, the Z-(DEVD)$_2$-R110 caspase substrate. One might expect $k(\phi)$ to be roughly constant since most enzymes operate far from the calculated Smoluchowski limit for diffusion control. However, $k(\phi)$ was found to decrease exponentially with increasing cytoplasmic concentration ($R^2 = 0.901$; Fig. 4f). We can therefore express the scaling of $k(\phi)$ as:

$$k(\phi) = k_0 e^{-g_k \phi}\tag{7}$$

where $k_0 \equiv k$ at $\phi = 0$ and $g_k$ is a scaling factor. These parameters were empirically estimated to be of 0.0022 $nM^{-1} \, min^{-1}$ and 0.531, respectively. We can then rewrite Luther's formula to explicitly include the three concentration dependencies:

$$v(\phi) = 2\sqrt{k_0 C_1 D_0 \phi e^{-(g_k + g_D)\phi}}\tag{8}$$

We can further simplify Eq. 8 by defining:

$$A \equiv k_0 C_1 D_0$$

$$g \equiv g_k + g_D$$

and arrive at a generalized version of Luther's formula:

$$v(\phi) = 2\sqrt{A\phi e^{-g\phi}}\tag{9}$$

where $A$, the speed factor, determines the magnitude of the trigger wave speed, and $g$, the cytoplasmic concentration scaling factor, determines how steeply the speed decreases at high cytoplasmic concentration. Using $A$ and $g$ as adjustable parameters, Eq. 9 fits well to the experimental data for apoptotic wave (Fig. 5a). We can also compare the fitted parameters to the quantities that contribute to them as estimated by experiments. Both $g_k$ and $g_D$ were individually measured (Fig. 4b, f), and their sum is close to the fitted value of $g$ (Fig. 5b). The product of the experimentally estimated values of $k_0$, $C_1$, and $D_0$ was also in reasonable agreement with the fitted value of $A$ (Fig. 5b), with the fitted value being about 19% lower than measured value of $k_0 C_1 D_0$. Given that actual value of $D_0$ could plausibly be up to 33% lower or up to 50% higher than that assumed here (see above), this seems like remarkably good agreement.

Note that Eq. 9 also predicts that at low cytoplasmic concentrations the wave speed should decrease, a trend that was not apparent in the initial experimental data (Fig. 5a). To test this prediction, we repeated the experiment over very low cytoplasmic concentrations, and, as shown in Fig. 5c, the low concentration data agreed well with curve fitting carried out on the higher concentration (0.5 x–2 x) results

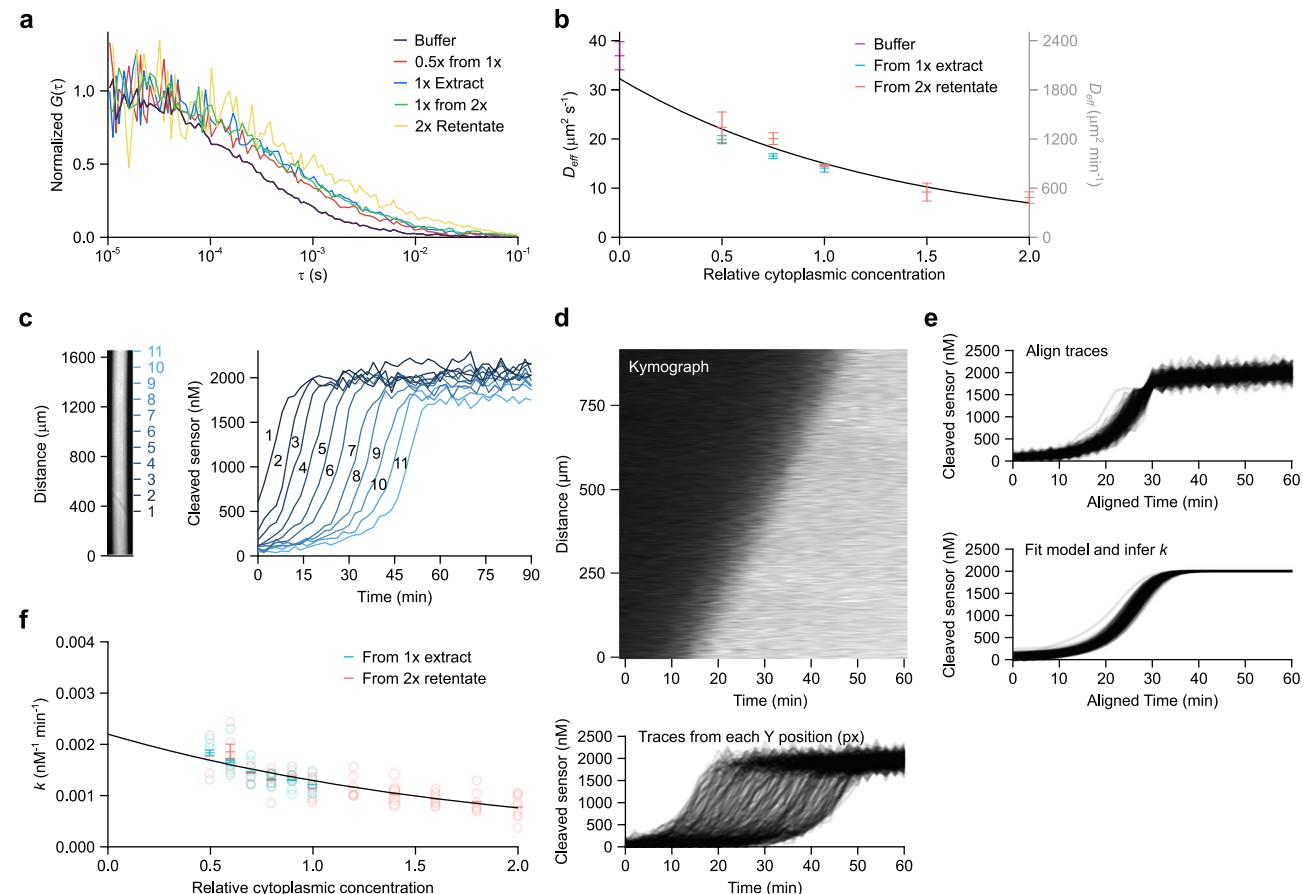

**Fig. 4 | The effective diffusion coefficient of AF488-BSA and apparent activation rate constant of caspase 3/7 decrease exponentially with cytoplasmic concentration. a** Representative fluorescence correlation spectroscopy (FCS) auto-correlation functions for AF488-BSA in extracts with different cytoplasmic concentrations. $G(\tau)$ is the autocorrelation function and $\tau$ is the time delay. **b** Effective diffusion coefficient of AF488-BSA at different cytoplasmic concentrations. Effective diffusion coefficients were calculated by fitting the auto-correlation data from FCS measurements to a Brownian diffusion model. A 60 s fluorescence intensity time course was registered for each FCS measurement. Means ± 90% CI calculated from 3 measurements ($n = 1$) are shown. Solid black curve is an exponential curve fitted to the means. **c** (Z-DEVD)$_2$-R110 cleavage kinetics can be monitored as apoptotic trigger waves sweep through a tube of extract. In this example, fluorescence from cleaved (Z-DEVD)$_2$-R110 at 11 positions (left) are shown on the right. Cleaved (Z-DEVD)$_2$-R110 concentration was inferred from fluorescence. **d** (Z-DEVD)$_2$-R110 cleavage kinetics in a kymograph (upper) can be represented as a series of traces (lower). **e** Traces shown in (**d**) were aligned by time (upper) and the model for caspase 3/7 activation and (Z-DEVD)$_2$-R110 cleavage was fitted to the data. Individual fitted traces are shown in the bottom panel. **f** The effective positive feedback rate constant $k$ at different cytoplasmic concentrations was extracted from the fitted model. Shown are means ± S.E.M. compiled from the same 9 experiments as the ones shown in Fig. 2d. The black solid curve is an exponential curve fitted to the means. Source data are provided as a Source Data file.

alone. Taken together, these findings show that combining Luther's formula for trigger wave speed and Phillies' equation for the concentration dependence of diffusion-limited enzyme activities yields an equation that accounts for the dependence of trigger wave speed on cytoplasmic concentration, including the near-maximal speed at 1 x concentration, the robustness of the trigger wave speed over a wide range of cytoplasmic concentrations, and the fall-off in speed at very high and very low concentrations.

Equation 9 could also be fitted well to the mitotic wave speed data (Fig. 5d). The fitted $g$ value was larger than that for apoptosis (1.54 vs. 1.34), which accounts for the observation that the wave speed fell more steeply with increasing cytoplasmic concentration.

### Mechanism of the robust apoptotic trigger wave speed

The robustness of the trigger wave speed appears to arise because one factor that influences wave speed, the concentration of the diffusible apoptotic mediator $C_{total}$, increases with increasing cytoplasmic concentration, whereas the positive feedback rate constant $k$ and diffusivity $D_{eff}$ decrease. Perfect robustness would arise if the competing trends canceled exactly. Here we examine in more detail how close to

exact the cancelation is and why it breaks down at very high and very low cytoplasmic concentrations.

We first expressed these functions in relative terms:

$$v_{rel}(\phi) \equiv \frac{v(\phi)}{v(\phi=1)},$$

$$C_{rel}(\phi) \equiv \frac{C_{total}(\phi)}{C_{total}(\phi=1)},$$

$$k_{rel}(\phi) \equiv \frac{k_C(\phi)}{k_C(\phi=1)},$$

$$D_{rel}(\phi) \equiv \frac{D_{eff}(\phi)}{D_{eff}(\phi=1)}.$$

Defining these relative quantities allows us to focus on the dependencies on cytoplasmic concentration $\phi$ while the proportionality constants $C_1$, $k_0$, and $D_0$ cancel out. We can also express Eq. 4 in

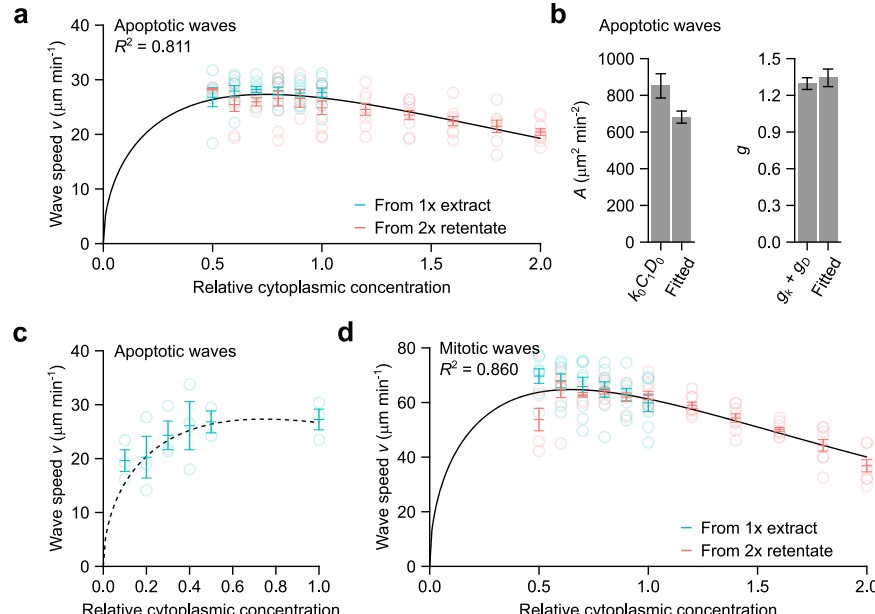

**Fig. 5 | Mitotic and apoptotic trigger wave speeds at different cytoplasmic concentrations follow the generalized Luther's formula. a** Apoptotic wave speeds at different cytoplasmic concentrations. The solid black curve is the generalized Luther's formula fitted to the means. Error bars are means ± S.E.M. Data are taken from the same 9 experiments as the ones shown in Fig. 2d. **b** The value of $A$, calculated as the product of the experimentally determined parameters $k_0$, $D_0$, and $C_1$, is compared to the fitted value (left panel). Likewise, the value of $g$ calculated as the sum of the experimentally determined parameters $g_k$ and $g_D$, is compared to the fitted value (right panel). $k_0$, $D_0$, $g_k$, and $g_D$ are from the exponential fits shown in Fig. 4. Error bars are S.E.M.s calculated directly from the fittings or propagated from the individual experimentally determined parameters. **c** Apoptotic wave speeds at low cytoplasmic concentrations. Dashed line shows the same fitted curve as in (**a**), which was obtained from only higher concentration data. Shown are means ± S.E.M. from 3 independent samples. **d** Mitotic wave speeds at different cytoplasmic concentrations, replotted from Fig. 1g and fitted to the generalized Luther's formula (black curve). Error bars are means ± S.E.M. Data are from 8 independent experiments. Source data are provided as a Source Data file.

relative terms:

$$v_{rel}(\phi) = \sqrt{k_{rel}(\phi) C_{rel}(\phi) D_{rel}(\phi)}, \tag{10}$$

If we take the logarithm, then the individual factors combine additively rather than multiplicatively:

$$\log v_{rel}(\phi) = \frac{1}{2}\left(\log k_{rel}(\phi) + \log C_{rel}(\phi) + \log D_{rel}(\phi)\right), \tag{11}$$

A robust trigger wave speed means $v_{rel}(\phi)$ should be close to 1 for a range of cytoplasmic concentration $\phi$. With log-transformation applied, a robust speed should have $\log v_{rel}(\phi)$ close to 0, meaning that the right-hand side of Eq. 11 should also be close to 0:

$$\log k_{rel}(\phi) + \log C_{rel}(\phi) + \log D_{rel}(\phi) \approx 0. \tag{12}$$

From 0.5x to 1x cytoplasmic concentration, the decrease in $\log D_{rel}(\phi)$ and $\log k_{rel}(\phi)$ combined (blue and green bars, respectively) is nearly equal to the gain in $\log C_{rel}(\phi)$ (red bars, plotted as negative to aid visual comparison; Fig. 6a). This explains why the apoptotic wave speed is almost constant over that range of concentrations. At higher cytoplasmic concentrations, the negative factors ($\log D_{rel}(\phi)$ and $\log k_{rel}(\phi)$) are larger in magnitude than the positive factor ($\log C_{rel}(\phi)$), and so the wave speed decreases with increasing cytoplasmic concentration, and at very low cytoplasmic concentrations, the opposite is true (Fig. 6a). Thus, the robustness of the trigger wave speed arises from the precise balancing of opposing kinetic and biophysical quantities over a range of cytoplasmic concentrations.

## Artificially maintaining diffusivity abrogates trigger wave speed robustness

One strong prediction is that if we could dilute an extract without increasing its diffusivity, the robustness of the trigger wave speed would be compromised. Toward this end we tested two viscogens, sucrose and BSA, and determined what concentrations would yield buffer solutions with diffusion coefficients equal to those seen in 1x cytoplasm. Using FCS and AF488-BSA, we found that 0.8 M sucrose (Fig. 6b) and 150 mg mL⁻¹ BSA (Fig. 6e) yielded diffusivities equivalent to that of 1x cytoplasm. We then diluted 1x cytoplasmic extracts with these buffers and asked whether trigger wave speed was no longer robustly maintained. As shown in Fig. 6c, f, trigger wave speed was now dependent upon cytoplasmic concentration over this range. Intermediate concentrations of the viscogens yielded intermediate wave speed results (Fig. 6d, g). Thus the exact balancing of the effects of cytoplasmic diffusivity and reactant concentration is the basis for the robustness in trigger wave speed.

## Discussion

Over the past several years it has become increasing clear that trigger waves are a recurring theme in both intracellular[8,12] and intercellular[16–18] communication. Unlike diffusive spread, trigger waves allow signals to propagate without diminishing in amplitude or slowing in speed. Trigger waves are complex, systems-level phenomena; they require biological reactions that include positive feedback loops, plus a spatial coupling mechanism. Any system as complicated as this is bound to have vulnerabilities. Here we have examined how vulnerable two intracellular trigger waves, apoptotic waves and mitotic waves, are to variation in the cytoplasmic concentration, a basic cellular property that differs from cell type to cell type, and even varies in individual cells as they proceed through mitosis. We found that even though a priori one might expect that a bimolecular reaction's speed would vary as the

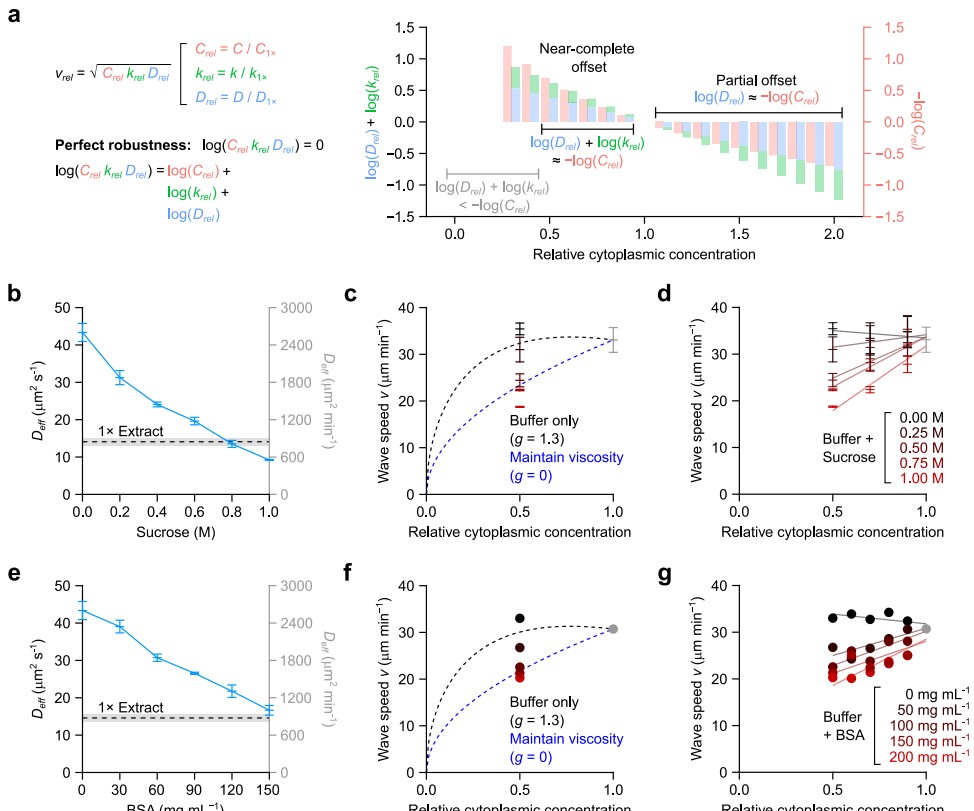

**Fig. 6 | Opposing kinetic and physical effects give rise to robustness in trigger wave speeds. a** Comparing the effect sizes of changes in rate constant, effective diffusion coefficient, and concentration. Effect size is defined as fold change relative to 1 x cytoplasmic concentration. For simplicity, the data for cytoplasmic concentrations < 0.3x, where the concentration effect dominates, are omitted. **b, e** Effective diffusion coefficients of AF488-BSA in XB buffers of various sucrose concentrations (**b**; no BSA present) or BSA concentrations (**e**; no sucrose present) as determined by FCS. Means ± 90% CI ($n = 1$) calculated from 3 measurements are shown. AF488-BSA diffusion in 1 x extract can be mimicked by ~0.8 M sucrose or ~150 mg mL$^{-1}$ BSA. **c, f** Apoptotic trigger wave speeds with extracts diluted with buffer (black data points) or a viscogen (red data points) and compared with theoretical curves (dashed lines). Crowding effects are quantified by the parameter $g$. For apoptotic trigger waves, $g$ is -1.3 (Fig. 5b) and is 0 if crowding effects are absent. We note that, in the case of $g = 0$, wave speed $v$ is proportional to the square root of

total caspase 3/7 concentration $\sqrt{C_{rel}}$. Means ± S.E.M. ($n = 3$) are shown for sucrose-containing buffers (**c**). Means ($n = 2$) are shown for BSA-containing buffers (**f**). The curves were set to equal the measured wave speeds at 1 x cytoplasmic concentration for both sucrose-containing and BSA-containing buffers. Apoptotic wave speeds at 0.5 x cytoplasmic concentration are also plotted. We note the scaling can be approximated by a horizontal line for $g = 1.3$ between 0.5 x and 1 x cytoplasmic extract, whereas for $g = 0$, a straight line with a positive slope. **d, g** Apoptotic trigger wave speeds are plotted for a range of sucrose-containing (**d**) or BSA-containing buffers (**g**) at different cytoplasmic concentrations. Means ± S.E.M. ($n = 3$) are shown for sucrose-containing buffers (**d**) whereas means ($n = 2$) are shown for BSA-containing buffers (**g**). Straight lines were fitted to each sucrose (**d**) or BSA (**g**) concentration. Only buffer without sucrose or BSA manifest straight lines with slightly negative slopes. Viscogen-containing buffers, be it sucrose or BSA, manifest positive slopes. Source data are provided as a Source Data file.

square of the cytoplasmic concentration, both apoptotic and mitotic wave speeds were nearly constant when extracts were diluted down from 1x to lower concentrations, and fell modestly at higher-than-physiological concentrations. We derived a simple model that accounts for both the robustness of trigger wave speed and the slowing seen at very high and very low cytoplasmic concentrations. The model implies that the robustness arises from canceling effects of cytoplasmic concentration: increasing cytoplasmic concentration should increase the speed by increasing the concentrations of the reactants, but should also decrease the speed by increasing viscosity and hence slowing both the local coupling process (diffusion) and the rate constants for the positive feedback reactions. This implies that if one were to change cytoplasmic concentration without changing viscosity, by diluting cytoplasm with buffer supplemented with the appropriate concentration of a viscogen, the wave speed should cease to be invariant, and indeed this was found to be the case. For large cells, such as the *Xenopus* eggs, the robustness of the mitotic wave speed could contribute to the reliability of the extremely rapid embryonic cell cycles in face of the physical stresses expected when an oocyte proceeds from the isotonic environment of the ovary to the hypotonic environment of the pond.

As mentioned above, recently it has been shown that some biological signals propagate as intercellular trigger waves in cell culture and in living tissues. Our theoretical framework may apply to tissue-level signal relay with appropriate generalization, with an intercellular process taking the place of intracellular diffusion as the local coupling mechanism.

This work adds to our burgeoning understanding of how physical properties of the cytoplasm constrain the operation of fundamental cellular processes. The present work also highlights the power of the *Xenopus* egg extract for studying the emergent regulatory functions that come from the differential responses of complex, coupled physical and biochemical processes.

We relied on several approximations to access the analytical power of the generalized Luther's formula. Experimentally, we approximated diffusivity for caspase 3/7 and the mitotic machines (CDK1/CycB and PP2A/B55 complexes) with AF488-BSA. Based on size, we may slightly over- or underestimate the scaling factor $g_D$, respectively. While BSA (66 kDa) diffusion appears to approximate that of caspase 3/7 (~60 kDa) reasonably well, it may underestimate effect of crowding of the diffusive spread of mitotic activity because CDK1/CycB/Cks1 (~8.5 nm diameter) and PP2A/B55 holoenzymes (~10 nm diameter) are up to ~1.2 x and ~1.5 x

larger than BSA (~7 nm diameter). This is consistent with the larger fitted $g$ for mitosis than for apoptosis. However, we have not experimentally determined $g_k$ for mitotic waves in this study, which could help us determine $g_D$ for mitotic waves and understand the extent to which crowding affects the mitotic control system. FRET-based CDK1 activity sensor have been developed and have now been shown to be able to resolve mitotic wave dynamics in *Drosophila* embryos and *Xenopus* extracts[35,48,49]. With careful calibration, the CDK1 sensor may be a powerful tool to determine the parameters in generalized Luther's formula for mitotic waves. We approximated anomalous diffusion with effective Brownian diffusion at a short distance range. Depending on the true length scale of diffusive mixing in trigger wave propagation, we may slightly over- or underestimate protein mobility. Model-wise, we approximated the dynamics of the bistable switches with logistic functions. We were not able to resolve the exact steps at which apoptosis or mitosis are limited by molecular crowding. Despite these limitations, our prediction error for apoptotic trigger wave speed was around 20% from the measured values, suggesting a good overall approximation. Better experimental approximation and further developments in the theoretical treatments for anomalous diffusion and traveling waves in bistable media should improve the prediction accuracy and provide more detailed understandings to this topic.

## Methods

### Preparation of *Xenopus* sperm chromatin and egg extracts

All animal work adhered to relevant national and international guidelines and received approval from the Stanford University Administrative Panel on Laboratory Animal Care (APLAC protocol 13307). *Xenopus laevis* animals (females, LM00531, males, LM00715, NASCO) used in this study were >3 years old. We prepared demembranated *Xenopus* sperm chromatin as described previously[50]. We typically included sperm chromatin in experiments with cycling extracts, using them as pacemakers for mitotic waves, at ~100 sperm per μL extract. We prepared *Xenopus* egg extracts following previously established procedures[51] with the following modifications. We collected freshly laid eggs and removed the jelly coats with 20 mg mL⁻¹ L-cysteine aqueous solution (pH 7.8) in under 5 min. We then quickly washed the eggs three times in 0.2x Marc's modified Ringer's solution (MMR; 20 mM NaCl, 400 μM KCl, 400 μM CaCl₂, 200 μM MgCl₂, 1 mM HEPES, 20 μM EDTA, pH 7.8). For cycling extracts, the eggs were activated with 0.5 μg mL⁻¹ calcium ionophore A23187 (C7522, Sigma) in 0.2 x MMR prior to packing and crushing. The A23187-containing buffer was promptly removed upon egg activation. We omitted this step for interphase-arrested extracts. We then washed eggs washed two times with crushing buffer (50 mM sucrose, 100 mM KCl, 100 μM CaCl₂, 1 mM MgCl₂, 10 mM HEPES-KOH, pH 7.7) before we packed the eggs through low-speed centrifugation (200 $g$ for 1 min, followed by 600 $g$ for 30 s). After packing, we then removed excess buffer to minimize dilution. For cycling extracts, we only crushed the eggs 20 min after activation to ensure meiotic exit was completed. The eggs were crushed by centrifugation at 16,000 $g$ at 4 °C for 15 min. We collected the cytoplasmic fraction of the crude *Xenopus* egg extract and kept on ice. Peptidase inhibitors leupeptin (L2023, Sigma), pepstatin (P5318, Sigma), chymostatin (C7268, Sigma), and actin polymerization inhibitor cytochalasin B (C6762, Sigma) were added into the extracts at 10 μg mL⁻¹ each. For interphase-arrested extract, we also added cycloheximide (01810, Sigma) at 100 μg mL⁻¹ to suppress cyclin B synthesis and the cell cycle. No cycloheximide was added to the cycling extracts. We centrifuged the extracts at 16,000 $g$ at 4 °C for 5 min one or two times to eliminate impurities before experiments.

### Dilution, concentration, and reconstitution of *Xenopus* egg extracts

We concentrated the extracts using a 10 kDa molecular weight cut-off centrifugal filter (UFC501096, Millipore). We achieved a 2 x

concentration by centrifuging three times for 10 min at 4 °C (a total of 30 min). We homogenized the extracts between each spin by gentle pipetting. We used XB buffer without sucrose (100 mM KCl, 100 μM CaCl₂, 1 mM MgCl₂, 10 mM HEPES-KOH, pH 7.7) as the primary diluent for adjusting cytoplasmic concentration. In experiments involving viscogens, sucrose or BSA were titrated in XB buffer without sucrose to the required concentrations from stocks. Stock solutions of sucrose and BSA were prepared at the same salt concentration as the basal XB buffer. Dilution and reconstitution were carried out by mixing the extracts with diluents to reach the desired volume fraction. We determined the protein concentrations of the extracts and the filtrate using the Bradford method (500-0006, Bio-Rad).

### Imaging apoptotic and mitotic trigger waves

To monitor apoptotic and mitotic trigger waves, we mixed extracts with biosensors and filled PTFE tubes (~100 μm inner diameter) for subsequent time-lapse fluorescent microscopy. We used interphase-arrested extracts for apoptotic trigger wave propagation. We monitored caspase 3/7 activity using (Z-DEVD)₂-R110 (13430, AAT Bioquest) at 2 μM unless otherwise specified. To enhance the signal-to-noise ratio, we included 200 μM MG-132 (S2619, Selleckchem) to suppress (Z-DEVD)₂-R110 cleavage through the proteasomes. After filling the tubes, we let the tubes stand at room temperature for 30 min before inducing apoptosis. We initiated apoptosis by dipping the tube in freshly prepared apoptotic extracts. Apoptotic extracts was prepared by adding 2 μM cytochrome c (C2867, Sigma) to fresh extract and incubating at room temperature for 30 min. The initiated tubes were then placed into custom-made imaging chambers, submerged in heavy mineral oil, and imaged at 2-min intervals at room temperature. To monitor mitotic trigger waves, we used cycling extracts. We used 200 nM SiR-tubulin (CY-SC002, Cytoskeleton, Inc) to visualize the microtubule networks, which dissolve at the onset of mitosis, providing a sensitive readout for the mitotic wave fronts. After filling of the PTFE tubes (~100 or ~300 μm in diameter), we immersed the tubes in heavy mineral oil in the imaging chambers and imaged at 2–3-min intervals at room temperature. We define biological replicates as extracts prepared from different clutches of eggs obtained from different females. Within each biological replicate, up to 5 tubes were monitored as technical replicates. Typically, 1 to 2 waves per technical replicate were observed for apoptotic wave experiments, and >2 waves per technical replicate were typical for mitotic wave experiments.

### Measurement of trigger wave speeds

We made kymographs from time-lapse videos of the propagating trigger waves in the PTFE tubes, using the Multi Kymograph program in FIJI/ImageJ. Kymographs were binarized or scaled to facilitate visual inspection. We manually fitted straight lines to the linear segments of the propagating waves and determined the wave speeds from the slopes of the fitted lines. Each biological replicate included up to five technical replicates. A median speed was initially calculated for each technical replicates, and then a median was calculated from the medians of the technical replicates.

### Measurement of protein diffusivity by FCS in *Xenopus* egg extract

FCS measurements in *Xenopus* egg extract were analyzed following a previously described method[42]. Briefly, we prepared and adjusted the cytoplasmic concentration of interphase-arrested extracts as described. We added the EDTA-free cOmplete protease inhibitor cocktail at a 1:50 (v/v) ratio, 30 min prior to the addition of 25 nM Alexa Fluor 488 labeled BSA (AF488-BSA) to the extracts. FCS data were acquired using an inverted Zeiss LSM 780 multiphoton laser scanning confocal microscope at room temperature (22 °C). The microscope setup and the calibration step were described previously[42].

The confocal spot was focused 30 – 40 μm above the dish surface. Each data point represented the average of at least 3 randomly selected positions within the extract field. At each position, fluorescence intensities were acquired for 60 s. Autocorrelation functions were calculated directly by the ZEN 2.3 SP1 FP3 310 (Black) software (version 14, Zeiss) controlling the microscope. An anomalous diffusion model or a Brownian diffusion model were used to fit the autocorrelation functions:

$$G(\tau) = \frac{1}{N\left(1+\left(\frac{\tau}{\tau_D}\right)^{\alpha}\right)\sqrt{\left(1+\frac{1}{s^2}\left(\frac{\tau}{\tau_D}\right)^{\alpha}\right)}}, \tag{13}$$

where $G(\tau)$ is the autocorrelation function, $\alpha$ is the diffusion-mode parameter as defined by the mean squared displacement (MSD) equation $MSD(t) \propto t^{\alpha}$, where $\tau_D$ is the characteristic diffusion time, $N$ is to the particle number, and $s$ is for the structural parameter of the optics. The Brownian diffusion model is identical to the anomalous model except that $\alpha = 1$ is enforced. Consistent with prior reports[42], protein diffusion in *Xenopus* egg extract was weakly subdiffusive behavior and was better described by the anomalous diffusion model. Notwithstanding, an effective diffusion coefficient can be calculated from $D_{eff} = \langle r^2 \rangle / (4\tau_D)$ where $r$ is the radius of the confocal spot.

### Logistic dynamics of CDK1 activation in cycling *Xenopus* egg extract

Measurements of CDK1 activity in the cycling *Xenopus* egg extract were conducted previously by Pomerening et al.[41]. We analyzed the data points between mitotic onset and exit (60 to 90 min). The relative CDK1 activity was renormalized, and a logistic curve (Eq. 2) was fitted to the data within the selected time range up to the point when maximal activity was reached.

### Determination of caspase 3/7 activation kinetics

We inferred caspase 3/7 activation kinetics from the cleavage of (Z-DEVD)2-R110 (see Supplementary Table 1). We built a model for this inference as described below. We approximated caspase 3/7 activation kinetics with logistic growth, using the following closed form expression:

$$C(t) = \frac{C_{total}C_0}{C_0 + (C_{total} - C_0)e^{-kC_{total}t}}. \tag{14}$$

Here, $C$ is the concentration of active caspase 3/7, $C_{Total}$ is the total concentration of inactive and active caspase 3/7, $C_0$ is the caspase 3/7 concentration at $t = 0$, and $k$ is the effective second-order rate constant for caspase 3/7 activation. (Z-DEVD)$_2$-R110 is cleaved by active caspase 3/7 and nonspecific background activity to release fluorescent rhodamine 110 (R110):

$$\frac{dR}{dt} = k_R C(R_{total} - R) + k_{BG}(R_{total} - R), \tag{15}$$

$R$ is the cleaved, fluorescent R110 concentration, $R_{total}$ is the total concentration of (Z-DEVD)$_2$-R110 and cleaved R110, $k_R$ is the rate constant for caspase 3/7 cleaving (Z-DEVD)$_2$-R110, and $k_{BG}$ is the first-order rate constant for background cleavage. We found a solution to $R(t)$:

$$R(t) = R_{total} - (R_{total} - R_0)\left(\frac{C_0}{C_{total}}\left(e^{kC_{total}t} - 1\right) + 1\right)^{-\frac{k_R}{k}} e^{-k_{BG}t}, \tag{16}$$

with $R_0$ representing fluorescent R110 concentration at $t = 0$.

To reduce the number of fitting parameters in Eq. 16, we experimentally determined $k_R$ for cytoplasmic concentration ranging from 0.5 x–2x. We measured (Z-DEVD)$_2$-R110 cleavage rate in freshly prepared, fully apoptotic extracts. We induced apoptosis by treating interphase extracts with 2 μM cytochrome c followed by a 30 min incubation at room temperature to fully activate caspase 3/7. We then added 5 μM (Z-DEVD)$_2$-R110 to the apoptotic extracts, immediately and vigorously vortexed at high speed, and promptly recorded the fluorescence increase. Maximal cleavage rates were determined from the initial time points using linear fitting ($R^2 \geq 0.99$). The apparent second-order rate constants were then calculated from these maximal rates. We assumed that caspase 3/7 concentration is linearly proportional to the overall cytoplasmic concentration and is 200 nM in 1 x extracts and that a negligible decrease in (Z-DEVD)$_2$-R110 during the early time points.

We deduced caspase 3/7 activation kinetics in apoptotic wave propagation by applying Eq. 16 to the experimental data. We used the same kymographs here as those used for trigger wave speed measurements. We fitted a two-component Gaussian mixture model on the kymograph, obtaining an estimate of the R110 signal asymptote from the larger component of the mixture model. We corrected the signals for background and calibrated them to the nominal concentration of (Z-DEVD)$_2$-R110. The signal intensity at $t = 0$ determined $R_0$, the initial rhodamine 110 concentration. $C_0$, $k$, and $k_{BG}$ were determined by fitting.

To reduce the complexity in fitting, we aligned the trajectories of (Z-DEVD)$_2$-R110 cleavage over time at each position (pixels on the spatial dimension) on the kymographs. The trajectories were aligned by the time point at which they become 1000 times more likely to have reached the asymptote then not as determined by the two-component Gaussian mixture model fit. A time window of 60 min around this time point was selected for fitting for each trajectory. We then fitted Eq. 16 to the R110 concentration within this time window. We used the SAE-MIX package in R for the fitting, which uses a mixed-effect model approach. The mixed-effect model decomposes a "population" of varied observations into a population-level mean effect and individual-level random effects. We reported the population-level $k$ and only considered individual-level $k$ when demonstrating the individual-level fit. For each biological replicate, we calculated the median value from its technical replicates.

### Statistics and reproducibility

Experiments measuring trigger wave speeds from 0.5 x–2 x concentrations were repeated with 8 or more independent samples. Experiments testing specific conditions, such as inclusion of viscogens or quantifying cell cycle durations, were repeated with 2 or more independent samples. FCS experiments included 1 biological sample. No experiment was excluded from the analysis. No statistical power calculations were used to predetermine the sample size and the investigators were not blinded to sample allocation during experiments and outcome assessment.

### Reporting summary

Further information on research design is available in the Nature Portfolio Reporting Summary linked to this article.

## Data availability

The authors declare that the data supporting the findings of this study are available within the paper and its supplementary information files. Source data are provided with this paper.

## Code availability

Code for extracting the parameter $k$ for the generalized Luther's formula used in this study is available online on Github [https://github.com/johsihuang/Robust_Trigger_Wave_Speed]; Huang, J.-H. (2024). Robust trigger wave speed in *Xenopus* cytoplasmic extracts (Version 1.0.0).

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

## Acknowledgements
We thank the members of the Ferrell lab for helpful comments on the manuscript. This work was supported by a grant from the NIH (R35 GM131792) to J.E.F. and (K99 GM143481) to W.Y.C.H.

## Author contributions
J.H., Y.C., and J.E.F. conceptualized the study. J.H., Y.C., and S.T. performed the extract experiments. W.Y.C.H. and J.H. performed FCS experiments. J.H., Y. C., and J.E.F. analyzed the data. J.H. and J.E.F. conceptualized the model. J.H. and J.E.F. wrote the manuscript. J.E.F. supervised the study. J.E.F. and W.Y.C.H. secured the funding.

## Competing interests
The authors declare no competing interests.
