## [Peer Review File · Nature Communications]

Robust trigger wave speed in *Xenopus* cytoplasmic extractsREVIEWER COMMENTS

Reviewer #1 (Remarks to the Author):

This manuscript addresses the dependency of trigger wave speed on cytoplasmic concentration. Signaling waves are emerging as a common mechanism of regulation in complex biological systems, thus understanding how such waves are controlled is very important. The authors take advantage of an in vitro system, the *Xenopus* extract, to manipulate cellular density which is interesting, as density appears to change in mitosis, when cells undergo senescence and could possibly be developmentally regulated. Thus, this paper addresses an important question. The observation that both mitotic and apoptotic wave speeds have a small dependency on cellular concentration is both interesting and well-supported by the data. To rationalize the observation, the authors perform a theoretical analysis using Fisher-Kolmogorov waves. Unfortunately, this part is very problematic and, in my opinion, needs to be fully reworked, as I suggest below. Collectively, I am excited about this paper but I find that the theory part has serious flaws.

Major point:

The authors approximate the behavior of the mitotic and apoptotic signaling networks with logistic equations that leads to a FKPP reaction-diffusion equation and thus FKPP waves. This is a problem. FKPP waves are unstable waves and not metastable waves. While the authors use interchangeably the term trigger waves for both FKPP waves and waves that they propose to arise from bistability, the equivalency is not justified. It is a well-known theoretical result (see reviews on this topic by Van Saarloos or a recent review by Di Talia and Vergassola on biological waves) that metastable waves have a well-defined velocity which is independent of noise, but that the same is not true for FKPP waves, which are in fact very sensitive to noise. Using a class of waves whose speed is sensitive to noise to argue for speed robustness seems problematic to me. Moreover, the speed of waves in FKPP is dependent on the geometry of the wavefront and the speed could be significantly faster than what the authors derive for shallow wavefronts (again the distinction of "pushed" vs "pulled" waves in FKPP waves has been the subject of extensive studies).

In summary, I find that the authors use the wrong model for "trigger (metastable) waves". Given the importance of the experimental findings, this paper will be read by many people interested in learning about waves who might not have the background to know that the use of FKPP waves is problematic in this context. So, I feel that it is essential (both for intellectual rigor and as a service to the field) that the authors amend this part of the paper. My suggestion would be that they derive equation [5] from dimensional analysis (really Luther's formula is nothing other than that) instead of using the FKPP equation. Then, most of the remaining analysis would be justified. Alternatively, if the authors want to use an analytically solvable model, they can use bistable systems with piece-wise linear term or a third order polynomial which are solvable analytically (see classic paper from Ben-Jacobs or the Murray's Book Mathematical Biology), although the expressions they would get would be more complicated and less intuitive.

Minor points:

1. Many non-linear relationships can be well-approximated by logistic dynamics, so that is not a strong argument in support of mechanism.
2. The authors might want to comment on the fact that in the first cell cycles the mitotic waves are not clearly visible but they become apparent over time.
3. Recent work in both *Drosophila* and *Xenopus* has shown that Cdk1 activity can be directly measured using FRET biosensors (work from Di Talia lab and Yang lab). It seems appropriate to mention these results if the authors intend to mention the limitations of this study, especially since it seems that it is now possible to measure Cdk1 activity in *Xenopus* extracts.

Reviewer #2 (Remarks to the Author):

In this manuscript, the author characterize the robustness of multiple types of trigger waves to variation in cytoplasmic density. The corresponding author previously discovered and characterized phenomenon of signal propagation through traveling waves in Cdk1 dependent mitotic entry and

Caspase 3/7 dependent apoptosis, using cell-free xenopus extracts. Additionally they have published two recent interesting papers on the topic of how cytoplasm density and self-organization into micro-compartments influences protein diffusion and reaction rates.

In the current work, they turn to the question of whether and how cytoplasmic density influences the velocity and period of trigger waves. Similar to their previous study they use a spin column to concentrate extract and then measure wave characteristic for extract at various levels of dilution inside a PTFE tube. They first determine that Cdk1 dependent trigger waves governing mitotic entry show a dependence on cytoplasmic concentration. Cycle period increases along with wave speed as the extract is diluted. Shortest period occurs at highest cytoplasmic concentration, which also produces the slowest wave speed. The authors then seek to explain this phenomenon over the course of their study. Additionally they demonstrate that that apoptotic signals generated from a cytochrome C containing extract point source generates waves whose speed is largely invariant of dilution. By measuring protein diffusion using FCS they demonstrate that proteins are less diffusive in higher concentrations of cytoplasm while also have a reduced autocatalytic rate constant. Finally they demonstrate that at extreme dilutions wave speeds slows. To explain the robustness of trigger wave speeds, the authors argue that opposing kinetic and physics effect give balance one another. At higher viscosities and lower cytoplasm concentration, wave speed drops. However as cytoplasm concentration increases, this also increases viscosity and reduces diffusion.

This beautiful study is significant because it explains both how waves can travel across cells with extreme dimensions, such as 1 mm diameter oocyte, and how wave propagation robustness is maintained in spite of changes to cytoplasmic dilution. Varying cytoplasmic density is a common occurrence during the cell cycle, in different spatial subregions of the cell, result of osmotic stress, and coupled to cell aging. Therefore evolving mechanisms to maintain signal robustness is ideal. Strengths of the work include careful experimental analyses and mathematical modeling to explain the source of robustness in trigger wave speeds. I have only a few suggestions for improvement

1. Cdk1 trigger wave period increases as cytoplasm is diluted. In xenopus blastomeres, cell cycle period increases as cells reach a threshold size or NC ratio. Is there any evidence that trigger wave period may be responsible for this effect. Have they measured wave speed and period as a function of sperm DNA:cytoplasm ratio in their extracts?

2. Use of BSA for diffusion coefficient calculations: the authors acknowledge that FCS measurements of BSA may be inaccurate for various protein components in these signaling circuits. Can they provide examples of MWs of various complexes and the extent to which this would alter their conclusions on robustness. What percent error is expected for Cdk1/CyclinB and PP1/PP2A complexes that have different MWs from BSA? The authors state "Depending on the true length scale of diffusive mixing in trigger wave propagation, we may slightly over- or underestimate protein mobility". They state error is 20% for apoptosis wave speed. Can they also provide bounds on this for Cdk1 trigger wave speed and period? Make estimates for largest and smallest proteins in the network and model how this effects propagation speed and period. Indicate upper and lower bounds.

Reviewer #3 (Remarks to the Author):

This is a very elegant study in which the authors investigate robustness of mitotic and apoptotic waves in the frog egg extract. After experimentally establishing that the wave speed does not depend on the protein density in the cytoplasm, they introduce a simple autocatalytic-logistic model for the reaction part of the propagating wave and experimentally confirm the validity of the model. By adding diffusion, they obtain the Fisher-Kolmogorov equation, for which there is a known expression of the wave speed: square root from the product of certain total concentration, reaction rate and diffusion coefficient. Reasonable assumptions and clever measurements provide cytoplasmic concentration-dependence of

these factors and of the wave speed. Because the diffusion coefficient and reaction rate turn out to be decreasing functions of the concentration, while the the total concentration of a key molecule is an icreasing functions of the concentration, the resulting speed is largely insensitive to the cytoplasmic concentration. The non-trivial predictions - that at low cytoplasmic concentration the speed drops, and at high cytoplasmic concentration the speed slowly decreases - are experimentally confirmed.

As I wrote, this is a very elegant study, novel, interesting, a pleasure to read. Modeling part is perfect. I am not an expert on the experiment, but the experimental part looks fine to me.

We thank the reviewers for their careful reading of the manuscript and their thoughtful comments. We have revised the manuscript in accordance with all of the reviewers' suggestions and describe these changes below:

Reviewer #1 (Remarks to the Author):

This manuscript addresses the dependency of trigger wave speed on cytoplasmic concentration. Signaling waves are emerging as a common mechanism of regulation in complex biological systems, thus understanding how such waves are controlled is very important. The authors take advantage of an in vitro system, the *Xenopus* extract, to manipulate cellular density which is interesting, as density appears to change in mitosis, when cells undergo senescence and could possibly be developmentally regulated. Thus, this paper addresses an important question. The observation that both mitotic and apoptotic wave speeds have a small dependency on cellular concentration is both interesting and well-supported by the data. To rationalize the observation, the authors perform a theoretical analysis using Fisher-Kolmogorov waves. Unfortunately, this part is very problematic and, in my opinion, needs to be fully reworked, as I suggest below. Collectively, I am excited about this paper but I find that the theory part has serious flaws.

Major point:

The authors approximate the behavior of the mitotic and apoptotic signaling networks with logistic equations that leads to a FKPP reaction-diffusion equation and thus FKPP waves. This is a problem. FKPP waves are unstable waves and not metastable waves. While the authors use interchangeably the term trigger waves for both FKPP waves and waves that they propose to arise from bistability, the equivalency is not justified. It is a well-known theoretical result (see reviews on this topic by Van Saarloos or a recent review by Di Talia and Vergassola on biological waves) that metastable waves have a well-defined velocity which is independent of noise, but that the same is not true for FKPP waves, which are in fact very sensitive to noise. Using a class of waves whose speed is sensitive to noise to argue for speed robustness seems problematic to me. Moreover, the speed of waves in FKPP is dependent on the geometry of the wavefront and the speed could be significantly faster than what the authors derive for shallow wavefronts (again the distinction of "pushed" vs "pulled" waves in FKPP waves has been the subject of extensive studies).

In summary, I find that the authors use the wrong model for "trigger (metastable) waves". Given the importance of the experimental findings, this paper will be read by many people interested in learning about waves who might not have the background to know that the use of FKPP waves is problematic in this context. So, I feel that it is essential (both for intellectual rigor and as a service to the field) that the authors amend this part of the paper. My suggestion would be that they derive equation [5] from

dimensional analysis (really Luther's formula is nothing other than that) instead of using the FKPP equation. Then, most of the remaining analysis would be justified. Alternatively, if the authors want to use an analytically solvable model, they can use bistable systems with piece-wise linear term or a third order polynomial which are solvable analytically (see classic paper from Ben-Jacobs or the Murray's Book Mathematical Biology), although the expressions they would get would be more complicated and less intuitive.

This is a good point: in general the wave front in FKPP waves is unstable, and drifts away from its nominal position in the presence of perturbations, whereas mitotic waves and apoptotic waves propagate with quite a constant speed and an apparently stable wave front. As the reviewer suggests, we have changed the theory to start from Luther's formula as an empirical description of how the speed of a trigger wave depends upon D and the speed of the positive feedback, rather than from the FKPP model. The new derivation runs from lines 186-288. We have also removed all statements suggesting an equivalence between trigger waves and Fisher waves.

Minor points:

1. Many non-linear relationships can be well-approximated by logistic dynamics, so that is not a strong argument in support of mechanism.

Right. We have changed the text to say simply that we find the overall dynamics to be well-approximated by the logistic equation, rather than saying the logistic dynamics imply the sort of mechanism assumed in Fisher's model. This is now discussed on lines 195-215 and is referred to in the Methods section (lines 557-560).

2. The authors might want to comment on the fact that in the first cell cycles the mitotic waves are not clearly visible but they become apparent over time.

As suggested, we have added a statement to this effect on lines 136-137.

3. Recent work in both *Drosophila* and *Xenopus* has shown that Cdk1 activity can be directly measured using FRET biosensors (work from Di Talia lab and Yang lab). It seems appropriate to mention these results if the authors intend to mention the limitations of this study, especially since it seems that it is now possible to measure Cdk1 activity in *Xenopus* extracts.

As suggested, we now mention Di Talia's and Yang's FRET biosensors in the Discussion (lines 423-425). We get very robust signals with SiR-tubulin and so have relied on this probe for the present work.

Reviewer #2 (Remarks to the Author):

In this manuscript, the author characterizes the robustness of multiple types of trigger waves to variation in cytoplasmic density. The corresponding author previously discovered and characterized the phenomenon of signal propagation through traveling waves in Cdk1 dependent mitotic entry and Caspase 3/7 dependent apoptosis, using cell-free xenopus extracts. Additionally they have published two recent interesting papers on the topic of how cytoplasm density and self-organization into micro-compartments influences protein diffusion and reaction rates.

In the current work, they turn to the question of whether and how cytoplasmic density influences the velocity and period of trigger waves. Similar to their previous study they use a spin column to concentrate extract and then measure wave characteristics for extract at various levels of dilution inside a PTFE tube. They first determine that Cdk1 dependent trigger waves governing mitotic entry show a dependence on cytoplasmic concentration. Cycle period increases along with wave speed as the extract is diluted. Shortest period occurs at highest cytoplasmic concentration, which also produces the slowest wave speed. The authors then seek to explain this phenomenon over the course of their study. Additionally they demonstrate that apoptotic signals generated from a cytochrome C containing extract point source generates waves whose speed is largely invariant of dilution. By measuring protein diffusion using FCS they demonstrate that proteins are less diffusive in higher concentrations of cytoplasm while also having a reduced autocatalytic rate constant. Finally they demonstrate that at extreme dilutions wave speeds slow. To explain the robustness of trigger wave speeds, the authors argue that opposing kinetic and physics effects give balance one another. At higher viscosities and lower cytoplasm concentration, wave speed drops. However as cytoplasm concentration increases, this also increases viscosity and reduces diffusion.

This beautiful study is significant because it explains both how waves can travel across cells with extreme dimensions, such as 1 mm diameter oocyte, and how wave propagation robustness is maintained in spite of changes to cytoplasmic dilution. Varying cytoplasmic density is a common occurrence during the cell cycle, in different spatial subregions of the cell, result of osmotic stress, and coupled to cell aging. Therefore evolving mechanisms to maintain signal robustness is ideal. Strengths of the work include careful experimental analyses and mathematical modeling to explain the source of robustness in trigger wave speeds. I have only a few suggestions for improvement

1. Cdk1 trigger wave period increases as cytoplasm is diluted. In xenopus blastomeres, cell cycle period increases as cells reach a threshold size or NC ratio. Is there any evidence that trigger wave period may be responsible for this effect. Have they measured wave speed and period as a function of sperm DNA:cytoplasm ratio in their extracts?

Very interesting question. We have carried out preliminary experiments along these lines. If you let an extract cycle for a sufficiently long time, the cell cycle period eventually begins to lengthen (up to ~2 or 3x) and the trigger wave speed slows. It is possible that this simply reflects the depletion of something hard to maintain in extract—ATP levels, for example—but it is also possible that this is related to the midblastula transition, being triggered by the density of replicating nuclei. We are pursuing this further, but it goes beyond the scope of the present paper.

2. Use of BSA for diffusion coefficient calculations: the authors acknowledge that FCS measurements of BSA may be inaccurate for various protein components in these signaling circuits. Can they provide examples of MWs of various complexes and the extent to which this would alter their conclusions on robustness. What percent error is expected for Cdk1/CyclinB and PP1/PP2A complexes that have different MWs from BSA? The authors state “Depending on the true length scale of diffusive mixing in trigger wave propagation, we may slightly over- or underestimate protein mobility”. They state error is 20% for apoptosis wave speed. Can they also provide bounds on this for Cdk1 trigger wave speed and period? Make estimates for largest and smallest proteins in the network and model how this effects propagation speed and period. Indicate upper and lower bounds.

If the molecular weight of a trigger wave mediator is, say, 3x that of BSA, by the Stokes-Einstein relationship the diffusion coefficient (D_0) should be $\sqrt[3]{3} = 1.5$ -fold smaller (67%); if it is 1/3 that of BSA, it should be 1.5-fold larger (150%). Since the trigger wave speeds depend upon the square root of the diffusion coefficient, the estimated trigger wave speed could be 82% to 122% of the estimate using BSA’s measured diffusion coefficient.

The estimates we make of trigger wave speeds based on Luther’s formula, a nominal diffusion coefficient of $15 \mu\text{m}^2\text{s}^{-1}$, and the experimentally estimated values of κ for apoptosis and mitosis, differ from the experimentally-determined values by 2% and 17%—quite reasonable given the uncertainty in D_0 . This is now pointed out in the text (lines 217-224).

Likewise, the estimate from fitting that we obtain for $k_0 C_1 D_0$ for apoptosis is only 19% lower than that obtained from the measured values of k_0 and C_1 , and the assumed value of D_0 . Given that D_0 could plausibly be 33% lower than that of BSA, this is very reasonable agreement. This is noted now (lines 298-300). We do not have all of these parameters estimated for mitosis, though we do know from fitting that the g value (the scaling factor) is somewhat larger for mitotic waves than for apoptotic waves (estimated from fitting to be 1.54 vs. 1.34). This could be because the mitotic mediator is larger than albumin, but given that we have an incomplete picture of the relevant parameters, we would prefer not to discuss it in detail in the manuscript.

Reviewer #3 (Remarks to the Author):

This is a very elegant study in which the authors investigate robustness of mitotic and apoptotic waves in the frog egg extract. After experimentally establishing that the wave speed does not depend on the protein density in the cytoplasm, they introduce a simple autocatalytic-logistic model for the reaction part of the propagating wave and experimentally confirm the validity of the model. By adding diffusion, they obtain the Fisher-Kolmogorov equation, for which there is a known expression of the wave speed: square root from the product of certain total concentration, reaction rate and diffusion coefficient. Reasonable assumptions and clever measurements provide cytoplasmic concentration-dependence of these factors and of the wave speed. Because the diffusion coefficient and reaction rate turn out to be decreasing functions of the concentration, while the total concentration of a key molecule is an increasing function of the concentration, the resulting speed is largely insensitive to the cytoplasmic concentration. The non-trivial predictions - that at low cytoplasmic concentration the speed drops, and at high cytoplasmic concentration the speed slowly decreases - are experimentally confirmed.

As I wrote, this is a very elegant study, novel, interesting, a pleasure to read. Modeling part is perfect. I am not an expert on the experiment, but the experimental part looks fine to me.

Thank you! We have modified the modeling part because of Reviewer 1's concern that an FKPP-type model doesn't fit the observed behavior of mitotic and apoptotic waves—the stable wavefronts—as well as a trigger wave model would. But of course Luther's formula applies just as well to trigger waves as it does to Fisher waves, so hopefully the revised theory will still seem fine to Reviewer 3.

REVIEWERS' COMMENTS

Reviewer #1 (Remarks to the Author):

The authors have addressed my previous comment and significantly improved the paper.

Reviewer #2 (Remarks to the Author):

The revised manuscript is excellent and ready for publication. The authors have addressed my comments.

Reviewer #3 (Remarks to the Author):

I am satisfied with the revisions